# Understanding the Impact of AI Generated Content on Social Media: The Pixiv Case

## ABSTRACT

In the last two years, Artificial Intelligence Generated Content (AIGC) has received significant attention, leading to an anecdotal rise in the amount of AIGC being shared via social media platforms. The impact of AIGC and its implications are of key importance to social platforms, *e.g.,* regarding the implementation of policies, community formation, and algorithmic design. Yet, to date, we know little about how the arrival of AIGC has impacted the social media ecosystem. To fill this gap, we present a comprehensive study of *Pixiv*, an online community for artists who wish to share and receive feedback on their illustrations. Pixiv hosts over 100 million artistic submissions and receives more than 1 billion page views per month (as of 2023). Importantly, it allows both human and AI generated content to be uploaded. Exploiting this, we perform the first analysis of the impact that AIGC has had on the social media ecosystem, through the lens of Pixiv. Based on a dataset of 15.2 million posts (including 2.4 million AI-generated images), we measure the impact of AIGC on the Pixiv community, as well as the differences between AIGC and human-generated content in terms of content creation and consumption patterns. Our results offer key insight to how AIGC is changing the dynamics of social media platforms like Pixiv.

## CCS CONCEPTS

• **Human-centered computing → Empirical studies in collaborative and social computing**.

## KEYWORDS

Social Media, Generative AI, Empirical Study

## 1 INTRODUCTION

Artificial Intelligence Generated Content (AIGC) refers to content that is produced using generative AI techniques, rather than being authored by humans. In recent years, generative AI techniques have evolved significantly. Since, society has become intrigued by an array of content generation products, such as ChatGPT [3] for text and Midjourney [6] for images, particularly as they enable the automated creation of large volumes of content in a short period of time. Preliminary evidence suggests that this has led to an increase in the amount of AIGC content being shared via online social platforms [50], triggering debate regarding aesthetic [16, 26, 44], ethical [29, 32, 35, 39], and legal [14, 22, 39] issues. AI-generated

*Conference'17, July 2017, Washington, DC, USA*
© 2023 Association for Computing Machinery.
ACM ISBN 978-x-xxxx-xxxx-x/YY/MM...$15.00
https://doi.org/XXXXXXX.XXXXXXX

images, in particular, have garnered significant attention, due to their impressive ability to simulate photographs and art.

Whereas studies have been conducted to investigate people's perceptions of AI-generated art [42], the impact of AIGC on online social media platforms is yet to be studied. We argue that the exploration of AIGC on social media is crucial from three perspectives. (*i*) *Policies of Social Media Platforms:* AIGC offers convenience and speed, yet also holds the potential to amplify harmful narratives and drown out alternative opinions. Platforms need to comprehend the implications of AIGC and the trade-offs involved to implement the most appropriate guidelines and policies. (*ii*) *Community Formation:* Creators and consumers of AIGC may differ significantly from more traditional human-generated content communities. While including them in traditional creator communities may enhance diversity, it poses the risk of introducing issues that could be harmful to the community. For example, the scale at which AIGC can generate social media posts may lead to other members leaving the community [2]. (*iii*) *Algorithms and Engagement:* Social media platforms utilize algorithms to recommend content to users. By discerning the differences between AIGC and human-generated content, platforms can fine-tune their strategies, thereby ensuring that models are not negatively impacted by the volume of data created by AIGC. We therefore argue it is vital to better understand how AIGC impacts platforms and communities, and how the creation and consumption of AIGC differs from human-generated content. To explore these issues, we propose the following questions:

- **RQ1**: How has the arrival of AIGC impacted the content creation *ecosystem*, in terms of the volumes of content created and consumed over time, as well as the levels of user engagement and content themes?
- **RQ2**: How do individual AI and human content *creators* differ, particularly in terms of their productivity, profile information and interaction behavior?
- **RQ3**: How does consumer engagement differ across AI and human generated *content*?

To answer these questions, we focus on one exemplar platform: *Pixiv* [7], an online community focusing on artwork. It serves as a dedicated forum where artists can showcase their illustrations and receive feedback from consumers. As of 2023, Pixiv hosts over 100 million artworks and receives more than 1 billion page views per month [8]. One key innovation of Pixiv is its transparent mechanism for sharing AI-generated images. These images are publicly tagged, making it easy for viewers (and researchers) to distinguish between AI- and human-generated content. We argue that this makes Pixiv an ideal lens through which to study the impact of AIGC. As far as we know, there is currently no other image-based social media at scale that offers similar features. To answer our RQs, we therefore gather a Pixiv dataset consisting of 15.2M artworks and over 937K creators. To the best of our knowledge, this is the first in-depth empirical investigation into the characteristics of AIGC on social platforms. Our findings include:

(1) The introduction of AIGC on Pixiv has resulted in a 50% increase in new artworks, but no corresponding increase in the number of views or user comments. (§4.1).

(2) After the introduction of AIGC, there has been a growth in the number of new creators, but a 4.3% decrease in newly registered creators of human-generated content. (§4.2).

(3) The themes and subjects of artworks have undergone significant changes, with a decrease in diversity and a higher concentration of adult content and female characters. (§4.3).

(4) Even though AI creators can generate artworks much faster, they do not upload significantly more artworks than human creators (only 2 artworks more in the 50th percentile). (§5.1).

(5) AI creators are less communicative, and are more focused on using their posts for monetization. (§5.2).

(6) Although AI-generated artworks are more popular in the middle and lower percentiles, they cannot match the most popular human-generated artworks. This indicates that AIGC plays an important role in content consumption, but is not yet able to replace the top-level human creators. (§6.1).

(7) Whereas human-generated content consumption centers around the most popular creators and themes (they receive more views/bookmarks per capita), the popularity of AI-generated content is more uniformly distributed. (§6.2)

(8) Both human and AI creators are more likely to interact within their respective groups, especially for human generated artworks, which receives 5x more comments than AI generated ones from human creators on average. (§6.3)

## 2 PRIMER ON PIXIV

**Overview.** Pixiv is a Japanese website and online community for artists, which emerged in Tokyo in 2007. In 2023, the platform hosts over 100 million artistic submissions with a weekly increase of 0.2 million based on our dataset (§3). It receives more than 1 billion page views per month [8]. The primary objective of Pixiv is to furnish artists with a platform to showcase their illustrations and receive feedback and user comments. Pixiv excludes most forms of photography, and the majority of artworks on the platform consist of *original* artworks inspired by Japanese anime, and manga. This makes it rather distinct from other platforms such as Instagram. The website relies on a comprehensive tagging system to categorize the myriad of artworks, serving as the foundation of the entire site. Pixiv also allows the posting of explicit sexual content. Due to this, the site employs filters that can be enabled by users.

**User Profiles.** A user's profile page on Pixiv provides an overview of the artist, including their self-reported nickname, birthday, gender, and location. Additionally, users have the option to share a brief biography about themselves. Furthermore, there is an additional profile section dedicated to showcasing the artist's workspace, offering insights into their creative environment.

**Artwork.** Creators submit their images to Pixiv as *Artworks*. An artwork on Pixiv can encompass multiple images. The creator of an artwork has the freedom to provide a title and compose a caption to accompany the images. Each artwork is accompanied by a set of up to 10 tags. Additionally, any user has the ability to view the number of views and bookmarks an artwork has garnered.

**Tags.** Free-text tags play a significant role in Pixiv as they facilitate the grouping of images based on shared themes and subjects. Each artwork is limited to a maximum of ten tags, and the creator of the artwork can designate certain tags as locked or unlocked. When a tag is unlocked, any user can modify or remove it in a crowd-sourced fashion. Users are also allowed to add additional tags, and there is a provision to report any tags that are deemed unpleasant. Although they are lots of common tags that are widely used [10], users are free to create completely new tags.

**Interaction.** Users on Pixiv can follow other users, allowing them to keep track of their artworks. Users can engage in direct communication by sending each other messages and have the option to add each other as friends, fostering a sense of community. Moreover, users can leave comments on images, with a character limit of 140.

**AIGC.** Pixiv now permits the uploading of AI-generated images and implemented new policies in late October 2022 [1]: (*i*) Artists are required to explicitly indicate whether their submissions are human-generated or AI-generated using a special toggle. This information is displayed when others view the artwork. (*ii*) Users have the option to filter out all AI-generated artworks if they prefer. This option is set to off by default. (*iii*) AI-generated artwork and human-generated artwork are ranked separately. We argue that this establishes Pixiv as a prominent platform for sharing AI-generated works, making it an ideal case study.

Further, in May 2023, Pixiv updated its policies to enforce stricter rules on AIGC, in response to the complains from human creators [9]. These changes include banning the use of AIGC to "emulate" a specific human creator's style, and completely prohibiting AIGC on Fanbox, a subscription-based platform linked to Pixiv that is a major source of monetization for Pixiv creators [18].

## 3 DATASET COLLECTION

To answer our RQs, we gather publicly available data on Pixiv. See Appendix §A (in supplementary material) for an ethics discussion.

**Artwork Data.** Each artwork has a unique ID that is chronologically sorted based on its creation time. Our crawling process covers the range of artwork IDs from 95180765 to 114914391, encompassing all the artworks created between 2022-01-01 and 2024-01-05. The dataset includes 15,203,948 artworks with 2,475,485 tagged as AI-generated (16.2%). We flag that before 2022-10, there may be a small number of AI-generated artworks that were not explicitly tagged. However, these are likely to be few in number as Stable Diffusion, which is known to have created the boost in AIGC on Pixiv, was released on 2022-08-22 and received wide uptake in 2022-10 (with the release of Stable Diffusion v1.5).

**Creator Data.** Every user is assigned a unique ID, and we gather user data for the creators of the artworks in our dataset. Our dataset comprises a total of 937,130 users who have submitted at least one artwork between 2022-01-01 and 2024-01-05. The data of a user also encompasses myPixiv (*i.e.,* friends) of the user and users being followed by the user. However, the data does not include the user's followers as it is not publicly accessible.

**Temporal Data.** To track the changes in comments, views, and bookmarks over time, we construct another dataset by performing daily crawls of data for artworks generated in a week. The

dataset consists of artworks within the ID range of 108171000 to 108395275, which corresponds to a one-week period from 2023-05-16 to 2023-05-24.

## 4 RQ1: ECOSYSTEM TEMPORAL ANALYSIS

In this section, we explore the impact of AIGC on the overall Pixiv ecosystem by inspecting the volumes of content creation, consumption and engagement both before and after the introduction of AIGC (**RQ1**).

### 4.1 Impact on User Activity

We first investigate the overall user activity within the platform, encompassing both content creation and consumption activity, as well as user interaction activity.

***Impact of AIGC on Creation Activity.*** Figure 1a illustrates the weekly count of new artworks, both before and after the policy change allowing AIGC. This represents the level of content creation activity on a weekly basis. We see that the number of new artworks per week has been consistently increasing since the introduction of generative AI until 2023-05, in contrast to the stable patterns seen beforehand. Between 2022-10 and 2023-05, the weekly peak rose from approximately 0.13 million to 0.2 million. This constitutes an increase of over 50%. After that, the figure depicts a significant 10% decline, attributed to the enforcement of more stringent regulations on AIGC as detailed in §2. That said, the number of human-generated artworks also decreased, which is a result of creators leaving Pixiv in protest against the perceived "abuse" of AI [2]. Overall, this suggests that the presence of AIGC has significantly influenced the overall activity level of content creation. Furthermore, the results in 2023-05 also indicate that without proper policies, the presence of AIGC can impact the overall activity of human-generated content creation.

***Impact of AIGC on Consumption Activity.*** We next explore if the temporal changes in consumption activity match the increase of creation activity caused by AIGC. We rely on two metrics for measuring the consumption activity of an artwork: the number of views and bookmarks. We first calculate the weekly proportion of AI-generated artworks. Then, we determine the proportion of views and bookmarks received by AI-generated uploaded in the same week. The results are plotted in Figure 1c. We see that the percentage of views garnered by AI-generated artworks is less than the percentage of AI-generated artworks, as of 2023, by approximately 8%. However, this is not mirrored in the case of bookmarks. Figure 1c highlights that AI-generated artworks contribute significantly to the overall tally of bookmarks, with 32% of bookmarks garnered by AI-generated artworks as of 2023-04.

The views reflect the general activity of consumption, while the bookmarks signifies specific interest in consumption. With this in mind, the results suggest that, although AI-generated artworks contribute a considerable proportion (20%) of the general attention, this level of consumption cannot match the level of creation. This might be attributed to the optional mechanism provided to users, allowing them to filter AIGC content, as discussed in §2. However, the proportion of bookmarks matches the count of artworks, indicating that there is a segment of users who do demonstrate a particular interest in AI-generated artworks.

***Impact of AIGC on Interaction Activity.*** Figure 1b displays a time series of the number of comments generated under the artworks uploaded each week. This provides a measure of the level of interaction received for human vs. AI generated content. It is evident that the number of comments remains consistent both before and after the introduction of AIGC. Furthermore, AI-generated artworks receive only a small fraction (12% as of 2023) of the total comments as presented in Figures 1c. Thus, although people appear interested in *viewing* AIGC, they do not proceed to *interact* with such users. The exact reason is unclear, yet we conjecture that consumers may feel that human creators are more connected to the artwork and therefore more worth interacting with.

### 4.2 Impact on Creators

We next investigate the temporal impact of AIGC on the engagement of creators, considering both the emergence of newly registered creators and the level of activity among existing creators.

***New Creators.*** Figure 1d illustrates a time series of the weekly count of new creators (determined by the time the user uploads their first artwork). In the figure, the classification of human-generated and AI-generated content is based on the user's artworks. If the user has uploaded both AI-generated and human-generated artworks, they will be classified into Mix. The results indicate an increasing trend in the number of new creators (rising from around 4K to 5K) following the introduction of AIGC. However, the period spanning from 2022-01 to 2022-04 witnessed the inception of 69,326 new human creators, whereas the timeframe from 2023-01 to 2023-04 saw a diminished count of 66,463 creators – marking a decrease of 4.3%. After the implementation of stricter policies on AIGC in 2023-05, the number of new creators rebounded. This may suggest that without strict rules on AIGC, the platform may become less appealing to new creators of human-generated content.

***Existing Creators Activity.*** We further explore the activity of existing creators. Figure 1e displays the weekly count of active creators categorized by the age of their accounts. The results indicate that the number of active "new" creators (with an account age of less than 1 year) has increased following the introduction of AIGC. However, the number of monthly active "old" users remains stable. It is notable that there are approximately 23,000 monthly active creators with accounts that are at least 5 years old (33% of the total monthly active creators). These suggest that the existing "old" creators continue to be as active as before.

Furthermore, Figure 1f illustrates the weekly count of artworks categorized by users belonging to different "age" groups (calculated as the age of the account). Here, we observe a similar trend among older creators, who sustain their previous levels of productivity. However, artworks from new creators also display a notable surge, comprising nearly 50% of the total artworks as in 2023-04. 56.2% of these artworks are AI-generated. This observation potentially implies that while the "old" creators retain their activity, their overall contribution to the ecosystem diminishes comparatively in the face of the burgeoning output from new AI creators.

### 4.3 Impact on Themes

We finally investigate the impact on the themes of the artworks. To identify themes, we rely on the Pixiv's tagging system. This

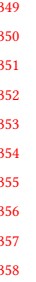
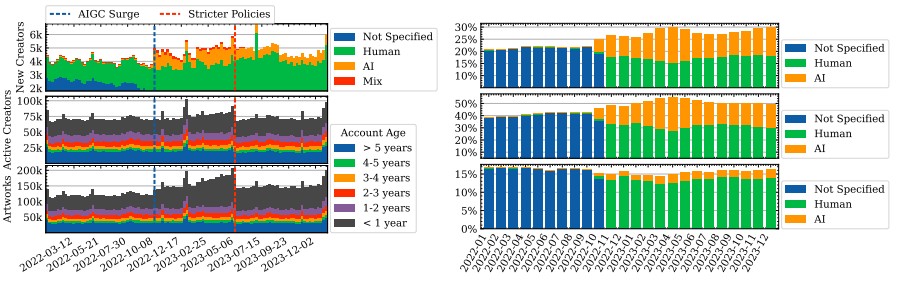

**Figure 1: Top to bottom: (a) Weekly count of new artworks; (b) Number of comments generated under the artworks uploaded on each week; (c) Weekly proportion of AI-generated artworks, and the proportion of views, bookmarks, and comments received by AI-generated artworks uploaded in the same week. (d) Weekly count of new creators; (e) Weekly count of active creators categorized by the age of the account; (f) Weekly count of artworks categorized by the age of the creator's account. (g) Proportion of restricted (adult) artworks; (h) Proportion of artworks with female-tags; (i) Proportion of artworks with male-tags.**

mechanism groups images based on shared themes and subjects, as discussed in §2 (see Appendix §B.1 in supplementary material for excluded tags).

***Diversity.*** As generative AI has the potential to enable individuals from all backgrounds and with varying interests to create an artwork, we first look at the diversity of themes within the artworks. We model content themes using the tags allocated to each artwork and measure diversity using the Gini index. For each month, we calculate the Gini index for tags on the number of artworks associated with tags. We observe an increase in the Gini index following the introduction of AIGC: it was 0.85 in `2022-10`, increasing to 0.90 by `2023-04` and 0.88 in `2023-12`. This outcome implies that the diversity of artwork themes has *declined* since the introduction of AIGC. To investigate this, we calculate the monthly ranking of tags (listed in Table 2 in the supplementary material). Through this ranking, we observe a notable increasing trend in tags related to R-18 content and female characters, while a decreasing trend in tags related to male characters. We therefore explore this trend in the following paragraphs.

***Restricted Content.*** Figure 1g illustrates the proportion of restricted (adult) artworks. The proportion of restricted artworks saw a significant increase after the introduction of AIGC, rising from approximately 20% per month to over 30%. As of `2023-12`, over half of all R-18 artworks are generated by AI, confirming that this increase can be attributed to AIGC. Overall, these findings indicate that the presence of AIGC has contributed to a substantial increase in adult content on Pixiv.

***Male vs. Female.*** In addition to the increase in restricted content, we also observe a new skew towards female characters. To investigate this trend, we group tags based on how frequent two tags are under the same artwork. We then manually label them as pertaining to male or female characters (see §B.3 in supplementary material for full details). Using this method, we obtain 439 representative tags for female characters (female-tags) and 98 representative tags for male characters (male-tags). Figures 1h and 1i illustrate the proportion of artworks with female-tags and male-tags, respectively.

Figures 1h shows a significant increase in the percentage of artworks featuring female-tags, rising from 41% to 53%. Furthermore,

a considerable proportion of these artworks are AI-generated, and this proportion has been increasing. As of `2023-04`, AI-generated artworks make up 25% of the total, nearly half of all female-tagged artworks. However, for male-tagged artworks displayed in Figure 1i, the percentage `keeps stable`, with only a small fraction being AI-generated. These results suggest that after the introduction of AIGC, the subject matter of artworks on Pixiv has shifted towards female characters, traditionally considered male-oriented themes.

***Diversity Revisit.*** To further explore the relationship between the previously observed decrease of diversity and R-18 and female characters content, we re-calculate the monthly Gini index, excluding both themes respectively, as shown in Figure 6 (in supplementary material). We observe that by `2023-04`, without R-18 artworks, the Gini index only increases from 0.83 to 0.86, which is smaller than the overall increase (0.85 to 0.9). Furthermore, when excluding artworks with female tags, the Gini index remains around 0.79 and does not show an increase. This finding confirms that the decline in diversity is associated with the rise in adult content and female characters in AIGC.

> **Takehomes:** (*i*) AIGC on Pixiv has led to 50% more new artworks, but no corresponding increase in views or comments. (*ii*) Without strict policies, AIGC can impact the engagement of creators, decreasing the number of newly registered human-generated content creators by 4.3%. (*iii*) Artwork themes have changed with less diversity and more adult content and female characters.

# 5 RQ2: PER-CREATOR ANALYSIS

Our temporal analysis in **RQ1** (§4) confirms that AIGC has had a significant impact on Pixiv. In order to gain a deeper understanding of AIGC, we now explore the difference between the *creators* of AIGC and human-generated content (**RQ2**). For our analysis, we utilize data from our dataset starting from `2022-11-01`, as this portion of the data is specifically categorized as either AI-generated or human-generated.

## 5.1 Creator Productivity

AIGC makes it possible to produce large volumes of content in a short period. Therefore, we first investigate the difference in productivity of AI-generated and human-generated creators.

***Artwork Creation Time.*** We first investigate the time taken by the creator to complete an artwork. We hypothesize that AI creators may produce content much faster. To estimate this, we calculate the interval (in days) between the upload times of two consecutive artworks by the same creator. The CDF is displayed in Figure 2a. It is evident that the interval between the upload times of AI-generated artworks are significantly smaller than those of human-generated artworks. 55% of AI-generated artworks are uploaded on the same day as the creator's previous work, whereas for human-generated artworks, this is only 20%. These findings suggest that the creation of AI-generated artworks takes significantly less time, confirming that AI creators are indeed more "efficient".

***Number of Artworks.*** We next proceed to investigate the number of artworks. We hypothesize that AI creators may produce much more artworks as we previously find that AI creators are more "efficient". Figure 2a plots the CDF of the number of artworks submitted by each creator. It is evident that AI creators have a larger number of artworks compared to human creators across all percentiles, albeit not significantly higher (as compared to the significantly higher "efficiency" of AI creators). At the 50th percentile, human creators have 4 artworks, while AI creators have 5 artworks. In order to gain a deeper understanding of the tail end of Figure 2b, Figure 2c plots the distribution of artwork uploaded by creators. We observe that the overall distribution of AI and human creators is similar, with a minority of highly productive creators generating the majority of artworks. Additionally, the figure shows that the contribution of the top AI and human creators is comparable, although the most productive AI creators contribute slightly more artworks in proportion (17% vs. 13%). Thus, in contrast to our previous observations, there is not a significant difference between human and AI creators in the number of artworks produced, either regarding the overall distribution or per-creator.

***Upload Patterns.*** The previous two paragraphs show that, although AI creators produce works faster (Figure 2a), the overall number of artworks produced by AI creators is not significantly higher than that of a human creator (Figure 2b). To better explain this, Figure 2d plots the count of active days of a creator *vs.* the count of their artworks. We see that AI creators generally require fewer active days to produce the same number of artworks. On average, an AI creator uploads 2.2 artworks per active day while this is 1.2 for a human creator. This is because many AI contributors upload a substantial batch of artworks at once, followed by periods of inactivity that span several days. This could be attributed to either AI creators (*i*) generating numerous artworks in a single session; or (*ii*) continuously producing images, but only uploading the finest pieces in bulk. The exact reason is unclear, but this process does set AI creators apart from their human counterparts.

## 5.2 Creator Profiles & Activities

We now examine the distinctions between the profiles, as well as the communication and monetization activities of AI creators and human creators. Note, the gender and job information are all self-reported, which may not be entirely reliable.

***Gender.*** Approximately 58% of human creators and 55% of AI creators do not provide their gender information in our dataset. Among the users who do disclose their gender information, approximately

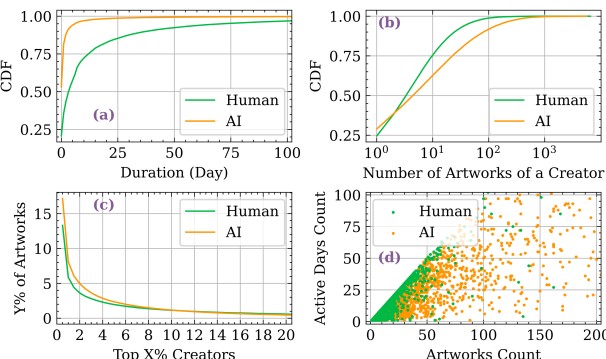

**Figure 2: (a) CDF of the difference between the upload times of two consecutive artworks by the same creator; (b) CDF of the number of artworks of each creator; (c) Distribution of artwork uploaded by creators; (d) Count of active days of a creator vs. the count of artworks created by the same creator.**

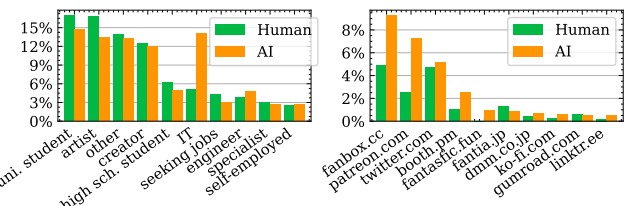

**Figure 3: (a) Jobs of creators; (b) Percentage of artworks that include a link to an external platform.**

50% of human creators identify themselves as male, and the other 50% identify as female. The AI creators stand in stark contrast, with a male:female ratio of 80-20. This observation may perhaps help explain the skew towards female characters discussed in §4.3.

***Job.*** Figure 3a presents the distribution of jobs among creators. The most significant observation is that around 14% of AI creators label themselves as working in the IT industry, which is notably higher than human creators (5%). Moreover, the percentage of AI creators working as engineers is also slightly higher compared to human creators with an engineering job (4.8% *vs.* 3.8%). This result is intuitive since AI creation requires certain skills that align well with individuals having IT backgrounds.

***Communications.*** We start by examining the communication behavior of the creators, as this is a crucial factor in fostering an active community. There are two broad ways that creators can communicate with their audience (beyond the image itself). These are by including text in the user profile, or by attaching a caption to the artwork. We therefore first focus on the presence of captions. We find that 42% of AI-generated artworks lack captions, which is much higher compared to human-generated artworks (29%). Additionally, 41% of AI creators do not have captions on their profile pages, while human creators only have 22% without captions. These findings indicate that AI creators might be less inclined to communicate personally with their audience.

***Monetization.*** We also notice that some users submit their artworks as samples or advertisements to promote their paid subscription on other platforms such as Fanbox, or to attract more followers on social media platforms such as Twitter. Indeed, we find that 57% of human creators and 28% of AI creators list a Twitter handle in their profile bio. To investigate this further, we extract all the external URLs in the captions of the artworks. We identify 9.72 million URLs in total, covering over 18.4k domains.

Figure 3b illustrates the percentage of artwork captions that include a link to the top 10 external domains. We find that 9% of AI-generated artworks include a link to Fanbox, surpassing the 5% found in human-generated artworks. Moreover, before 2023-05 (when Fanbox started banning AIGC), the numbers are 16.4% *vs.* 5%. Additionally, we also find that a higher percentage of AI-generated artworks contain links to other monetization platforms (*e.g.,* Patreon and booth) when compared to human-generated artworks. These findings suggest that AI creators are more inclined to promote their paid subscriptions. This also aligns with our results in §4.1: the prohibition of AIGC in Fanbox led to a significant decrease in the number of AI-generated artworks.

> **Takehomes:** (*i*) AI creators generate artworks faster but do not upload significantly more artworks than human creators. (*ii*) There are significant differences in gender ratio and jobs between AI creators and human creators. (*iii*) AI creators are less communicative and make greater use of monetization services.

## 6 RQ3: PER-CONTENT ANALYSIS

Through our investigation on **RQ2** (§5), we discovered a contrast between the producers of AIGC and human-generated content. We are next curious if there are also difference in the consumption of individual artworks. Therefore, we next investigate the levels of engagement received for AI vs. human generated content (**RQ3**).

### 6.1 Overall Consumption Activity

We begin by contrasting the overall consumption volumes for AI-generated vs. human-generated individual artworks. We rely on two metrics for measuring the consumption activity of an artwork: the number of views and bookmarks. For this, we utilize the temporal data covering views and bookmarks outlined in §3. Figure 4a displays the count of views and bookmarks in the 50th, 75th, 90th, and 99th percentiles in relation to the age of the artwork.

***Views & Bookmarks.*** We see that AI-generated artworks tend to have more views in the 50th and 75th percentile, but are surpassed by human-generated artworks in the 90th percentile, and lag even further behind in the 99th percentile. Specifically, in the 90th percentile, human-generated artworks have 33% more views, and in the 99th percentile, human-generated artworks have a remarkable 200% more views. The bookmarking outcomes for AI-generated artworks exhibit a more favorable trend. AI artworks tend to accumulate more bookmarks within the 50th, 75th, and 90th percentiles. However, human-generated artworks surpass AI-generated ones by a factor of 1x in the 99th percentile. The results suggest that, although AI-generated artworks are not yet able to compete with top-tier human-generated artworks, lower- and mid-popularity AI artworks seem to surpass human-generated material. Thus, whereas there is

no evidence that top-tier human creators are being crowded-out, there may be a more detrimental impact on amateur creators.

***Distribution Differences.*** The above suggests that human artworks may experience a more skewed popularity distribution. To confirm this, we measure the difference in the popularity distributions of human-generated vs. AI-generated content. Here, we calculate the Gini index for the number of views and bookmarks, respectively, for all artworks in our temporal data. For each artwork, we extract their view/bookmark count 10 weeks (70 days) after they were initially uploaded. The Gini index for human-generated artworks is 0.82 for views and 0.85 for bookmarks, whereas for AI-generated artworks, it is 0.64 for views and 0.69 for bookmarks. Confirming our intuition, AI-generated artworks *do* have a substantially more uniform popularity distribution. We suspect this is because the quality difference among human artworks is far greater than that among AI-generated content.

### 6.2 Popular Themes & Creators of Consumption

We are next curious about the distribution of views across creators and themes (using tags, see §B.1 in supplementary material for excluded tags). To gain deeper insights into this, we zoom in to one single day in our temporal data. Figure 4b present a boxplot of the number of views and bookmarks per artwork. For each artwork, we take their counts 10 weeks (70 days) after they were uploaded. We group the artworks by three categories respectively: (*i*) *None* presents the distribution of the number of views/bookmarks across all artworks, *i.e.,* grouped by nothing; (*ii*) *Tag* presents the distribution per-tag, *i.e.,* each data point represents the average number of views/bookmarks for all artworks associated with that tag; (*iii*) *Creator* presents the distribution of views/bookmarks per creator.

We see that the distribution of human-generated artworks is more varied than that of AI-generated works, with similar medians but significant differences in the 95th percentile and higher outliers. This suggests that the consumption of human-generated content tends to focus on the most popular artworks, creators, and themes. While for AI-generated content, the consumption is more evenly distributed. One possible explanation is that the quality of AI-generated artworks is more consistent, allowing even novice creators to produce reasonable works. In contrast, for human creators, it is natural that some artworks produced by talented artists are far superior to most other artworks and, therefore, attract a larger audience. However, this fails to explain the distribution of tags. Another possible explanation is that consumers of human-generated artworks tend to have specific targets, such as themes or creators, leading to a centralized distribution around popular tags and creators. In contrast, consumers of AI-generated artworks may be more likely to browse randomly.

### 6.3 Who Comments?

We now look at the comments left on artworks. This reflect the interaction activity of creators and consumers. We are curious about whether there is a difference between those who comment on AI- *vs.* human-generated artworks. To investigate this, "commenters" are classified into six categories: AI creators, human creators, mixed creators, unspecified creators, not creator, and self if the comment is from the creator themselves. Figure 5a displays the percentage

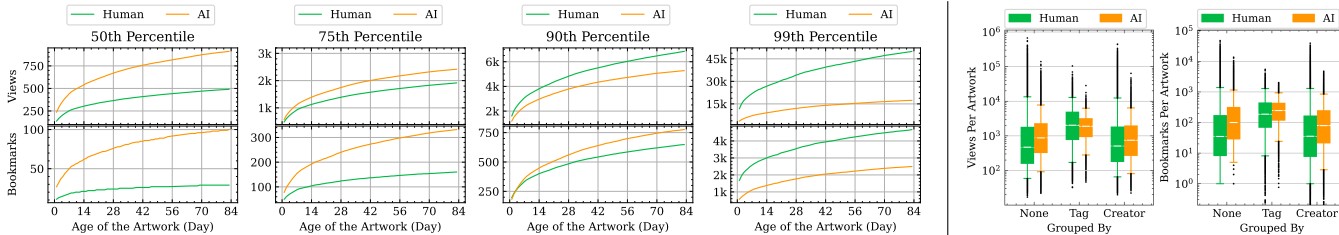

**Figure 4: (a) Count of views and bookmarks in the 50th, 75th, 90th, and 99th percentiles in relation to the age of the artwork; (b) Boxplot of the views and bookmarks per artwork, the whiskers are 5th and 95th percentiles.**

of comments under AI- and human-generated artworks from different user categories. Most comments (more then 80%) are from non-creators or the creators themselves. We also observe another interesting trend in which human/AI creators have more comments under human-/AI- generated artworks, respectively. To measure this, for each user category $C$, we define ratio:

$$R = \begin{cases} NC\_H^C / NC\_AI^C & \text{if } NC\_H^C / NC\_AI^C > 1 \\ -(NC\_H^C / NC\_AI^C)^{-1} & \text{otherwise} \end{cases}$$

where $NC\_H^C$ is the number of comments on human-generated artworks from $C$, and $NC\_AI^C$ is the number of comments on AI artworks from category $C$. A larger value indicates that users in this category have more comments under human-generated artworks. For example, $R = 2$ for the category ($C$) of all-users, indicates that from all-users, human-generated artworks receive twice as many comments compared to AI-generated artworks.

The results are plotted in Figure 5b. Indeed, we see that from all-users, the ratio is 8, indicating that human-generated artworks receive 8x as many comments as AI-generated artworks. Interesting, we also see homophily: From human-creators, human-generated artworks receive 15x more comments than AI-generated artworks. In contrast, from AI-creators, human-generated artworks receive just 0.4x more comments than AI-generated artworks. These trends, however, are partly driven by the fact that there are more human-generated artworks than AI ones (1.8x more in our dataset). Thus, we also normalize this ratio by the number of artworks. Specifically, we calculate normalizer $N = NA\_H / NA\_AI$, where $NA\_H$ is the number of human-generated artworks and $NA\_AI$ is the number of AI-generated artworks. Then we calculate the ratio $R' = NC\_H^C / NC\_AI^C / N$, then the normalized ratio $R_n$:

$$R_n = \begin{cases} R' & \text{if } R' > 1 \\ -(R')^{-1} & \text{otherwise} \end{cases}$$

This offers the ratio of the average number of comments received per artwork. For example, a normalized ratio of 2 from the category all-users, indicates that from all-users, human-generated artworks receive twice as many comments *on average per artwork* than AI-generated artworks do. The normalized ratios are also depicted in Figure 5b. We see that, when normalized by the number of artworks, AI-generated artworks actually obtain twice as many comments from AI-creators, compared to human-generated artworks. These results confirm that human creators are more likely to interact with human-generated artworks, and AI creators are more likely to interact with AI-generated artworks, suggesting a notable difference in the interaction between *creator* groups under artworks.

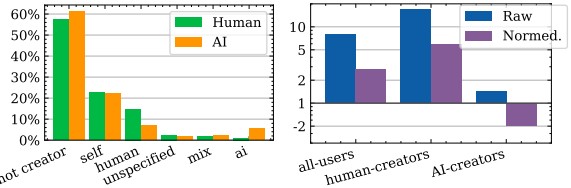

**Figure 5: (a) Category of the users who left the comments; (b) Ratio of the number of comments under human-generated and AI-generated artworks from different user categories.**

> **Takehomes:** (*i*) AI-generated artworks are more popular in the middle and lower percentiles, but cannot match the top human-generated ones. (*ii*) Human-generated content consumption centers around the most popular creators and themes, while for AI-generated content it is more evenly distributed. (*iii*) Both human and AI creators are more likely to interact within their respective groups.

## 7 DISCUSSION

Through our investigation on **RQ1** (§4), we confirmed that AIGC has had a significant impact on the overall ecosystem, particularly in terms of content creation volumes. Further, our analysis of **RQ2** (§5) and **RQ3** (§6) reveals notable differences between the creators and consumers of AIGC compared to human-generated content. These findings offer key insights into the role of AIGC within online social media. Here, we discuss important implications.

***Abuse of Generative AI.*** There are concerns that with the advent of generative AI, certain users may generate an abundance of low quality content and share it via social platforms. However, our study does not reveal such abuse. Although the allowance of AIGC on Pixiv has led to 50% more artworks (§4.1), this is arguably within a reasonable range. Furthermore, we find that the number of bookmarks received by AI-generated artworks can actually match this increase (§4.1), confirming these AI artworks are not spam. Further, we find that AI creators are more "efficient" (creating artworks faster), but the number of artworks produced by an AI creator is not significantly higher than that of a human creator (§5.1). This is because they tend to follow a specific pattern, where they upload in batches. This could be due to their inclination to produce multiple images in a single sitting or creating a large number of pieces and then sharing the best ones in batches. These findings could be

useful for platforms looking to develop policies and algorithms to better manage AI-generated content and their creators.

***Impact on Human Creators.*** There have been criticisms within the artistic communities regarding the influence of AIGC on the engagement and activity of creators of human-generated content [4, 5]. Indeed, our findings reveal a 4.3% decline in the number of newly registered human creators following the introduction of AIGC, lending support to this view (§4.2). Additionally, although our research in §4.2 indicates that the presence of AIGC has had a limited impact on active existing users and the quantity of new human-generated artworks, we also observe a significant increase in the number of artworks originating from new creators. These make up almost 50% of all artworks as of 2023-04, and most of these (56.2%) are generated by AI. Moreover, we find that AI artworks in the mid- and lower-percentiles seem to surpass human-generated ones (§6.1). This suggests that while the engagement levels of "old" creators remain the same, their significance in the community may be diminishing compared to the massive output of new (AI) creators. Although our results show that AI cannot compete with top-tier human creators, amateur human creators may suffer.

***Impact on Content Diversity.*** The impact of AIGC on the themes of the content is significant. We observe a decrease in the diversity of themes, as evidenced by the Gini index. This decrease can be attributed to a higher concentration of adult content (50% increase in proportion) and female characters (30% increase in proportion). Overall, our results suggest that the introduction of AIGC can profoundly alter the themes of the platform, thereby potentially influencing the culture of the community. This is perhaps one of the more worrying trends, and we argue that it is crucial for online social platforms to approach this matter with careful consideration.

***User Communication & Interaction.*** Our analysis of the creator profile reveals differences between AI and human creators concerning gender ratio and job (§5.2). Further, §5.2 highlights that AI creators tend to be less communicative and are more motivated by monetization. With time, this may impact the demographics and culture of the platform. The interaction on human- and AI-generated content also differs. Only a small proportion of comments are under AI-generated artworks (§4.1), suggesting that people might be more inspired to reach out for human created content. We argue that these findings have key implications for community formation. Online social communities should carefully consider how to incorporate AI creators, who may not actively engage in communication within the existing community.

***Content Consumption Behaviour.*** Our results reveal that AI-generated artworks are more popular in the lower percentiles (under 90th percentile), while human-generated artworks are more popular in the higher percentiles (§6.1). Additionally, AI-generated artworks get more bookmarks per view, indicating consumers may have specific interests in them. In contrast, consumption of human-generated content focuses on popular creators and themes, while for AI-generated content it is more uniformly distributed (§6.2). This may be due to the consistency in quality of AI-generated artworks, or simply because consumers of human-generated artworks tend to have specific targets of creators and tags. Either way, these findings highlight a significant difference in content consumption behavior between human-generated and AI-generated artworks.

This sheds important light on how people perceive AIGC. We also posit that this may offer useful insights into the design of recommendation algorithms, which may benefit from using the use of AI tags as an explicit feature.

## 8 RELATED WORK

***Image-based Social Media Platforms.*** Numerous studies have measured image-based social platforms. Among these platforms, Instagram has receives the most attention [43]. Other platforms that have been studied include Flickr [37, 40, 47], Pinterest [21, 33, 41], and Tumblr [13, 25]. These platforms are generally more focused on photographs and have a broader purpose. Additionally, there have been studies on platforms for artists, such as Behance [28], Dribbble [24], ArtStation [49], and deviantArt [12, 45, 49]. These platforms cater more to designers and their need for design elements. Thus, their art styles and communities are vastly different from that of Pixiv. While there have also been studies conducted on Pixiv [31, 48], they are qualitative and questionnaire-based. To the best of our knowledge, this is the first large-scale empirical study of Pixiv.

***AIGC on Social Media Platforms.*** Several recent studies have delved into the repercussions of AIGC on social media platforms. Haque et al. investigated the attitudes of ChatGPT users using Twitter data and found that early adopters expressed positive sentiments, especially in areas such as entertainment and creativity [23]. This correlation is consistent with our findings, which show an increasing number of AI-generated artworks on Pixiv and a high bookmark rate for them. Chen and Zou compiled a dataset of AI-generated images on Twitter, revealing a significant amount of NSFW, pornographic, and nude content [15]. Another study analyzed people's perceptions of ChatGPT using content from Twitter and Reddit, highlighting concerns about potentially harmful or NSFW content [19]. These findings align with our observations of the rise in R-18 content on Pixiv. Several other recent studies have reported similar results [30, 38, 46]. However, these investigations focus on individuals' attitudes and opinions toward AIGC, while our paper directly measures the impact of AIGC and its unique characteristics at scale.

Other studies have examined the use of AI-generated images in social media [20, 27, 34, 36]. They come from a security perspective though, *e.g.,* fake content and phishing. Additionally, there have been studies exploring the use of AIGC for journalism [11] and advertisements on social media [17]. Our research is distinct from previous studies since we measure the impact and characteristics of AIGC on social media based using a large-scale dataset. To the best of our knowledge, we are the first to undertake such a study.

## 9 CONCLUSION

This paper has presented a comprehensive study of the impact of AIGC on Pixiv, a major online community for sharing artworks. Our findings provide valuable insights into the effects of AIGC on online social media. We hope that our findings can support platforms in taking appropriate measures to address the challenges of AIGC and enhance their policy formulation, community construction, and algorithmic design. In the future, we hope to expand our analysis to other platforms, such as Twitter and Instagram.

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
