# OpenReview forum: "Understanding the Impact of AI-Generated Content on Social Media: The Pixiv Case"
_acmmm.org/ACMMM/2024/Conference — MM2024 Poster_

### Official Review · Reviewer_PVHS · 2024-05-11

**Rating:** 3
**Confidence:** 3

**Summary:**

This work conducts an empirical study on the impact of AI-generated content (AIGC) on the online art community platform Pixiv. By analyzing a large-scale dataset of over 15 million artworks, including 2.4 million AI-generated images, the authors investigate how the introduction of AIGC has influenced content creation, consumption, user engagement patterns, etc. The study reveals various interesting phenomena, such as changes in content themes, differences in creator productivity and interaction behaviors, and disparities in popularity distribution between human-generated and AI-generated artworks. The paper addresses an important and timely research question, as the rise of AIGC is a significant trend in online content creation and sharing. Understanding its impact on platforms is crucial for platform governance, community management, and algorithmic design.

**Strengths:**

1. The large dataset scale (over 15 million artworks) ensures the robustness and generalizability of the findings.
2. The paper comprehensively analyzes multiple aspects of AIGC's impact, providing a holistic understanding of the phenomenon.
3. The findings reveal several interesting and thought-provoking phenomena and show the complex dynamics between AIGC and online art communities.

**Limitations:**

However, there are several concerns I have after the reading:

1. The paper lacks important methodological details, such as defining and calculating key metrics like the "Gini index," which is crucial for readers to assess the validity and reliability of the analyses.

2. While the paper excels at describing "what" phenomena are observed, it has limitations in explaining the "why" behind them since many analyses and interpretations are speculative and lack rigorous verification. For example, the authors attribute AI creators' preference for monetization platforms to their profit-seeking motivation, which could be confounded by other factors, such as the difference between their platform reputations. In other words, the "old" creators who do not use AI tools might have a stable way to obtain rewards from their fans or advertisements, so they do not need to perform in this "profit-seeking" style. Similarly, the lower user interaction with AI-generated content is interpreted as users' preference for human creation, but it could be due to the sensitive nature of AI-generated content (as the authors described, more NSFW content) that discourages users from publicly engaging with it. To substantiate these claims, the paper would benefit from more sophisticated statistical analyses, such as panel data models or causal inference methods, to tease out the true underlying mechanisms.

I think maybe the authors could add a comparative analysis between the human creator and AI creator in the "New" creators in some dimensions. In this way, it might solve my concern.

3. The operationalization of some key concepts, such as creator productivity, lacks justification and connection to prior literature. The authors directly use metrics like "artwork creation time", "number of artworks", and "upload patterns" to represent productivity without explaining why these are valid measures and how they relate to productivity measures used in previous studies. Strengthening the conceptual rigor and grounding the study in existing literature (e.g., extending the current "related works" section) would enhance the paper's academic contribution.

**Suitability:**

3

---

### Official Review · Reviewer_gmqH · 2024-05-13

**Rating:** 4
**Confidence:** 3

**Summary:**

In this paper, the authors explore the impact of AIGC on online social media platforms such as Pixiv and the differences in content creation and consumption patterns between AIGC and manually generated content. Specifically, the following three questions are addressed: Question 1: How has the arrival of AIGC impacted the content creation ecosystem in terms of the amount of content created and consumed over time, as well as the level of user engagement and content topics? Question 2: How do AI and individual human content creators differ, particularly in terms of productivity, profile information, and interaction behavior? Question 3: How does consumer engagement differ between AI and human-generated content?

**Strengths:**

1. The outstanding contribution of this author lies in his extensive collection of a large amount of data on the Pixiv platform, and through exhaustive data analysis, he delves into the unique impact of AIGC on social media platforms such as Pixiv and its characteristics. His study not only compares the differences between AI and humans in terms of quantity, time, gender, type of work, and communication when creating works but also further reveals the different performances of AI-generated artworks and human creations in interactive sessions such as comment interactions.

2. The authors' analysis of AIGC's impact on Pixiv largely matches our human intuition. A striking finding, however, is that while we usually assume that AI is far faster than humans in terms of creation speed and deserves to produce more work, the amount of work created by AI is lower than the amount created by human artists. This point provides a new perspective and provokes deeper thinking about how AIGC operates in the social media environment.

3.  Strong experiment performance.

4. Well-structured paper.

**Limitations:**

1. In Section 4.3. Which subgraph in Figure 1 corresponds to 'Restricted Content'? Which subgraphs in Figure 1 correspond to (h) and (i)? The author needs to annotate them in Figure 1. Additionally, please check if the legend is correct, for example, whether 'male' and 'female' should be in the legend?"

2. Please carefully review the description in lines 728-731 of your paper to ensure there is no mistake.  Should they be described as human commenters and AI commenters, rather than human-generated works and AI-generated works?

3. This paper is less relevant to multimodal multimedia, but it is a paper that can help us understand the impact of AIGC with social media.

**Suitability:**

2

---

### Official Review · Reviewer_KQu3 · 2024-05-25

**Rating:** 3
**Confidence:** 3

**Summary:**

This research examines the influence of AI-generated content (AIGC) on the content creation landscape, comparing the distinctions between AI and human creators, as well as the variations in consumer engagement with AI versus human-produced content. The analysis is centered on Pixiv, an online platform where artists can share their illustrations and obtain feedback.

**Strengths:**

(1) The topic is timely and interesting.

(2) The manuscript is mostly easy to follow.

(3) It covers multiple aspects of social media dynamics, including content creation, consumption patterns, and interaction metrics, providing a relatively comprehensive view of AIGC's impact.

**Limitations:**

(1) This study presented a comprehensive data analysis; however, the research question was only examined on a single dataset from one platform. Consequently, the findings may lack generalizability and robustness. This limitation could make the work resemble a data analysis report more than a research paper.

(2) The figures were not well-captioned, making it unclear how the captions correspond to the sub-figures.

Minor limitations:

(1) The untagged AI-generated content before certain policy changes was not included which may introduce some bias.

(2) The paper could explore more about how individual creators are adapting or competing with the influx of AI-generated content beyond just engagement metrics.

**Suitability:**

3

---

### Meta-Review · Area_Chair_XFDZ · 2024-07-01

**Recommendation:** Accept (Poster)
**Confidence:** 4

**Metareview:**

The authors have studied the impact that AI-generated content has on a specific social media Pixiv. This is an online community for artists who share their paintings and illustrations in order to receive feedback. An important aspect of Pixiv is that it allows both human and AI generated content to be uploaded. Thus, while the authors have focused on a single platform, the dataset is huge and highly valuable for the designed study. Indeed, it contains 15.2 million posts, including 2.4 million AI-generated images. So, the critique about using a single dataset does not reduce the value of the findings. In particular, the work compares the differences between AI and humans in terms of several dimensions such as quantity, time, gender, type of work, and communication as well as in interactive sessions such as comment interactions.

Moreover, as noticed by one of the reviewer the experiment is well designed. In addition, after the rebuttal 2 out of three reviewers raised their scores. Finally, the authors' rebuttal provides a good argumentation on the "what" and "why" explanations. About this last point, authors should avoid single and too speculative explanations in the final version.